energy/mechanical engineering

marine diesel engine, exhaust gas recirculation, performance evaluation, optimal exhaust gas recirculation rate, optimization algorithm

**Author for correspondence:**
Chuan-lei Yang
e-mail: yangchuanlei@hrbeu.edu.cn

# Experimental study on diesel engine exhaust gas recirculation performance and optimum exhaust gas recirculation rate determination method

Xiang-huan Zu, Chuan-lei Yang, He-Chun Wang and Yin-yan Wang

College of Power and Energy Engineering, Harbin Engineering University, Harbin 150001, People's Republic of China

X-hZ, 0000-0002-2678-4750; C-lY, 0000-0001-7468-5370

In order to study the exhaust gas recirculation (EGR) performance of marine diesel engines, a venturi high-pressure EGR device was established to overcome the exhaust gas reflow problem based on a certain type of turbocharged diesel engine. The EGR performance test is accomplished and an optimal EGR decision-making optimization method based on grey correlation coefficient modified is proposed. The results show that the venturi tube EGR can basically meet the injection requirements of high-pressure exhaust gas and achieve good results. Through the venturi tube EGR, the $NO_X$ emissions reduce significantly with the maximum drop of 30.6%. The explosive pressure in cylinder reduces with the EGR rate increases and the cylinder pressure curve shows a single peak at low-speed conditions and double peaks at high-speed condition. However, the fuel consumption rate, $NO_X$ and smoke have been negatively affected. Due to small samples, the traditional evaluation method is difficult to determine the optimal EGR rate reasonably, while the proposed method can effectively solve this problem. It can weaken the shortcomings of subjective judgement and greatly improve the rationality of decision-making results.

## 1. Introduction

With the strengthening of people's environment protection consciousness, the emission regulations of marine diesel engine are becoming stricter and stricter. In particular, the International

Maritime Organization has established specific emission limitation for marine diesel $NO_X$ pollutants. Owing to increasingly strict emission regulation, almost all marine diesel engines are equipped with exhaust gas recirculation (EGR) to reduce $NO_X$ pollutant emissions effectively [1,2]. The focus of EGR technology is to overcome the 'exhaust gas backflow problem' and control the EGR rate rationally. Since the pressure of the intake air is much higher than EGR exhaust gas, the high-pressure exhaust gas cannot be introduced into the intake port normally. At present, the main implementation of EGR includes the installation of a throttle or a venturi tube [3,4]. Among them, the former is easy to implement, but it will have a certain negative impact on the turbocharger and deteriorate the scavenging effect of the diesel engine, thus affecting the normal operation of the diesel engine. The latter only has a small effect on the intake pressure while it is not conducive to the arrangement.

In addition to the implementation of EGR, there are many other researches about EGR, which include EGR modelling and simulation [5], the experimental performance study [6,7], the EGR control research [8,9] and so on. However, there is little relevant research on EGR performance assessment, and no uniform standards for determining the optimal EGR rate exist. It can be seen from the existing literature that the methods commonly used by different scholars are empirical methods based on experiment. For example, Han [10] proposed to achieve the best EGR rate by optimizing $NO_X$ emissions below 10% during peak smoke periods. Shuai *et al.* [11] proposed to achieve the best EGR rate according to the criterion that the particle emission of 13 working point not exceeds the original machine. Zhang [12] proposed to achieve the optimal EGR rate according to the criterion that the $NO_X$ emissions meet the Tier III standard. Zhang [13] proposed the optimal EGR rate should achieve the biggest decrease in effective fuel consumption. Other researchers such as Du [14] and Zhang *et al.* [15] adopted the similar method. The above methods are based on the professional analysis, and the optimal EGR rate can be obtained successfully by each different criteria. However, they have some common disadvantage that it is over-reliance on subjective judgements. Although it can make full use of the experience of experts and technicians, it is too subjective that the decision-making results always vary from person to person because of different emphases and choices of decision-makers. On the other hand, each method is dependent on the integrity of the data, but for some specific occasions with 'small samples or poor data', there is no way to get enough data due to test conditions restrictions, and these methods will no longer apply.

The optimal EGR rate problem is a multi-attribute decision-making problem. Considering the advantages of grey decision-making in this field, this paper introduces the multi-objective grey decision-making theory into the performance evaluation of EGR. However, due to the different requirements of EGR under different working conditions, it is unreasonable to use the traditional grey decision-making model directly. Although different scholars have proposed many optimization decision models, different models have different starting points and focuses, which are only suitable for specific problems [16–21]. Therefore, it is necessary to find an optimization method that specifically meets the actual characteristics of diesel engine EGR.

In this paper, a venturi tube high-pressure EGR system on TBD234V12 turbocharged diesel engine is established and the main EGR operating parameters were obtained through experiments. Combined with the EGR characteristics of different operating conditions, an optimization grey decision-making method is proposed, which uses a mathematical model of subjective and objective comprehensive optimization to achieve EGR performance evaluation for the first time and obtains the predictive effect no matter whether it is a large sample or a small sample. This proposed method uses an objective mathematical model to explore the intrinsic relationship between different EGR parameters so as to evaluate the advantages and disadvantages of different EGR schemes, which can effectively solve the problem of 'subjective dependence' in current EGR performance evaluation and make the decision-making result more reasonable and realistic.

# 2. Test design and implementation

## 2.1. Test equipment

The research object of this paper is TBD234V12 type turbocharged diesel engine, the basic parameters can be found in [22] and the main test instruments are shown in table 1.

## 2.2. Venturi tube EGR system design

As we all know, the intake pressure of diesel engine is significantly higher than the exhaust, which makes it difficult to achieve EGR. Considering the spatial layout of the diesel test bench, this paper adopts the

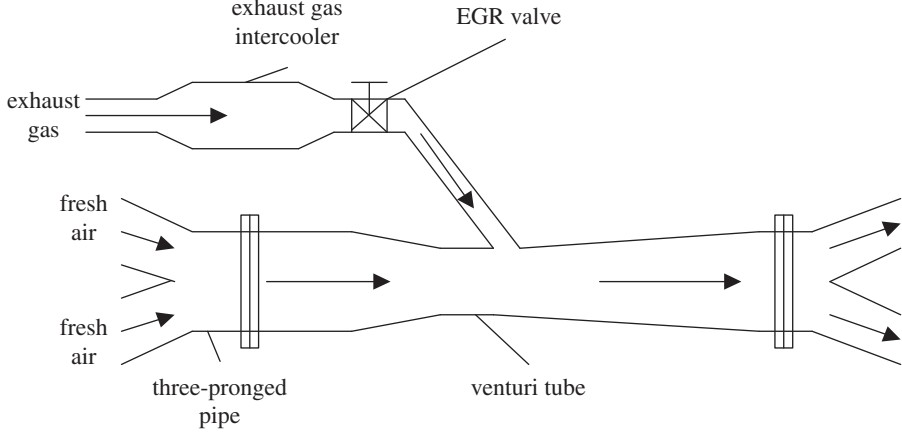

**Figure 1.** Venturi tube system structure diagram.

**Table 1.** Main instruments and equipment.

| test instrument | test parameters | measurement precision |
|---|---|---|
| AVL I60 exhaust analyser | $NO_X$, $CO_2$, CO, CH and so on | $\pm$ 1 ppm |
| Testo 350 flue gas analyser | intake $CO_2$ composition | $\pm$ 0.3% |
| 439 OPACIMETER opacity smoke meter | smoke | 0.001 $m^{-1}$ |
| FC2210 fuel consumption meter | fuel consumption rate | 0.1 g $kW^{-1}\,h^{-1}$ |
| SG880 hydraulic dynamometer | torque | 1 N m |

series venturi system to achieve the mixing of exhaust gas and intake air, and the schematic diagram is shown in figure 1. The fresh air compressed by two compressors in parallel is merged into a venturi inlet section through a three-pronged pipe, and then the air flows through the contraction section, the flow rate increases and the pressure decreases. After passing through the exhaust gas intercooler and the EGR valve, the exhaust gas is mixed with fresh air at the throat portion and the pressure is restored in diffuser section, and finally, the exhaust gas reaches each cylinder for combustion.

The main parameters that affecting the performance of the venturi are the throat area and the cone angle of the diffuser. The throat area determines the ejector capacity of the venturi and the cone angle of the diffuser determines the recovery of the gas pressure after mixing. To simplify the calculation, the flow is considered to be a constant flow situation, and the gas dynamics formula is applied [23],

$$\text{equation of state for ideal gas:} \quad p = \rho RT, \tag{2.1}$$

$$\text{flow continuous equation} \quad m = \rho Av, \tag{2.2}$$

$$\text{intake sound velocity equation} \quad a = \sqrt{\gamma RT} \tag{2.3}$$

and
$$\text{Mach number calculation equation} \; M = \frac{v}{a}, \tag{2.4}$$

where $p$ is the intake pressure, MPa; $\rho$ is the intake density, kg $m^{-3}$; $R$ is the gas constant, J $kg^{-1}\,K^{-1}$; $T$ is the absolute temperature of the intake air, K; $m$ is the intake air mass flow, kg $s^{-1}$; $A$ is the pipe cross-sectional area, $m^2$; $v$ is the intake air flow rate, m $s^{-1}$; $a$ is the local speed of sound, m $s^{-1}$; $\gamma$ is the specific heat ratio and $M$ is the Mach number.

In this paper, the rated working condition of the diesel engine (1800 r.p.m., 444 kW) was selected as the venturi tube design condition. The selection of this operating point is based on the following considerations: the flow in the pipe is simplified to constant flow, when the diesel engine is working in the design working condition and the opening degree of the EGR valve is adjusted from fully closed to fully open, the venturi can be started. By the action of the pressure-reducing ejector, the EGR rate required can be achieved, and when the EGR valve is fully opened, the throat portion of the

venturi tube will not occluded. Due to the limitation of the original machine structure, the medium-cold high-pressure exhaust gas circulation system is selected in this test.

The basic boundary conditions are determined as follows: the diameter of the inlet and outlet of the venturi tube should be equal to the diameter of the intake pipe $d_1 = d_2 = 115$ mm, the air pressure at the outlet of the compressor $p_1 = 0.157$ MPa, the temperature $T_1 = 345$ K, the flow rate of the intake air $m_1 = 0.529$ kg s$^{-1}$ and the pressure of the exhaust gas before the turbine $p_2 = 0.15$ MPa. According to the gas dynamics equation, it can be calculated

$$\text{intake density:} \quad \rho = \frac{p_1}{RT_1} = \frac{1.57 \times 10^5}{287.04 \times 345} = 1.5854 \text{ kg m}^{-3};$$

$$\text{intake flow rate:} \quad v_1 = \frac{m_1}{\rho A_1} = \frac{0.529}{1.5854 \times 0.115^2 \times 3.14/4} = 32.124 \text{ m s}^{-1};$$

$$\text{local sound speed:} \quad a_1 = \sqrt{\gamma RT_1} = \sqrt{1.4 \times 287.04 \times 345} = 372.344 \text{ m s}^{-1};$$

and
$$\text{Mach number:} \quad M_1 = \frac{v_1}{a_1} = \frac{32.124}{372.344} = 0.08627.$$

According to the Mach number, linear interpolation is used to check the gas dynamic function table

$$\frac{A_1}{A_*} = 6.7598 \quad \frac{p_1}{p_0} = 0.9948 \quad d_* = 44 \text{ mm}.$$

where $A_*$ is the critical section area, m$^3$; $p_0$ is the stagnation pressure, MPa.

The magnitude of the pre-turbine exhaust pressure determines the design value of the venturi throat pressure. In order to achieve a good ejector effect, it is necessary to form a certain pressure difference between the exhaust pipe and the throat portion of the venturi. According to empirical data, a pressure of 3–10 kPa is generally required [24]. As a result

$$p_t = (0.15 - 0.01) \times 10^6 = 0.14 \text{ MPa},$$

$$\frac{p_1}{p_t} = \frac{0.157}{0.14} = 1.1214$$

and
$$\frac{p_t}{p_0} = \frac{p_1/p_0}{p_1/p_t} = \frac{0.9948}{0.1214} = 0.8871.$$

According to the Mach number, linear interpolation is used to check the gas dynamic function table

$$M_2 = 0.41722, \quad \frac{A_t}{A_*} = 1.5372$$

$$d_t = \sqrt{\frac{A_t}{\pi/4}} = \sqrt{\frac{A_1}{\pi/4} \cdot \frac{A_t/A_*}{A_1/A_{cr}}} = \sqrt{\frac{1.5372}{6.7598}} \times 0.115 = 54.84 \text{ mm} > d_*.$$

Therefore, the design meets the requirements. After determining the throat area, the nozzle length of the venturi $L_1$, the length of the mixing section $L_t$ and the length of the diffuser $L_2$ are determined according to the empirical formula. In order to balance the space arrangement of the test bench, the selection of the total length $L$ must be feasible.

In this test, the shrinkage cone angle $\alpha = 24°$ which meets the empirical value $10° < \alpha < 40°$

$$L_1 = \frac{d_1 - d_t}{2tg(\alpha/2)} = \frac{0.115 - 0.05484}{2tg12°} = 0.1415 \text{ m} = 141.5 \text{ mm}.$$

Considering the overall size layout of the test bench, set $L = 441.5$ mm, $L_t = 50$ mm

$$L_2 = L - L_1 - L_t = 400 - 141.5 - 50 = 245 \text{ mm}.$$

Therefore, the diffuser angle

$$\beta = 2\arctan\frac{d_1 - d_t}{2 \times L_2} = 2\arctan\frac{0.115 - 0.05484}{2 \times 245} = 14°.$$

According to experience, the diffuser angle $\beta$ should be within the range $11° < \beta < 18°$ [11], so the designed diffuser section meets the requirements. The total calculated parameters are shown in table 2.

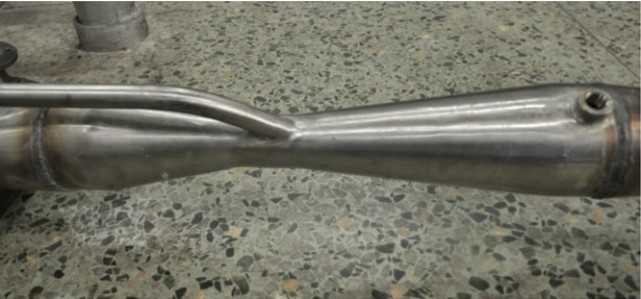

**Figure 2.** Venturi physical map.

**Table 2.** Venturi tube main parameters.

| the throat area | 54.84 mm | the length of the mixing section | 50 mm |
|---|---|---|---|
| shrink cone angle | 24° | the length of the diffuser | 245 mm |
| the length of the venturi nozzle | 141.5 mm | the diffuser angle | 14° |
| the total length | 441.5 mm | | |

The physical map of venturi is shown in figure 2.

# 3. System installation and experimental design

## 3.1. System installation

In order to avoid the damage to diesel engine caused by discarded particles, high-pressure EGR system is selected in this paper. The original test is a V-type diesel engine and in order to facilitate the EGR, the original double inlet pipe was transformed into a single inlet manifold and a bypass pipe was added to the two rows of exhaust pipes. As a result, the exhaust gas can be drawn from the bypass pipe to ensure the stability of the two turbochargers and the uniformity of exhaust gas that return to the cylinder.

Figure 3 shows the EGR system structure diagram; figure 4 shows the test bench physical map; figure 5 shows the Venturi tube installation diagram.

## 3.2. Testing programme

Due to the limitations of the test conditions, as well as to ensure the stable operation of the diesel engine, the maximum EGR rate is limited within 15%. The test consists of three loads (25%, 50% and 75% load) at three speeds (low, medium and high speed). The detail schemes are shown in table 3.

As the $CO_2$ tracer method is the most common and effective method for steady-state test currently which basically meet the accuracy requirements [17], the $CO_2$ tracing method is used to calculate the EGR rate in this paper, the $CO_2$ concentration in the intake and exhaust and the atmospheric environment are detected, respectively

$$\eta_{EGR} = \frac{\phi_{CO_{2in}} - \phi_{CO_2}}{\phi_{CO_{2out}}} \times 100\%.$$ (3.1)

where $\phi_{CO_{2in}}$ is the concentration of $CO_2$ in the intake manifold; $\phi_{CO_{2out}}$ is the concentration of $CO_2$ in the exhaust manifold; $\phi_{CO_2}$ is the concentration of $CO_2$ in the atmosphere.

# 4. Performance test results analysis

## 4.1. Influence of EGR rate on cylinder pressure

Figure 6 shows the influence curve of EGR on cylinder pressure under different operating conditions. It can be seen from figure 6 that the variation trend of cylinder pressure curve under the same working condition is basically similar. A single peak appears at low and medium speeds and the peak pressure in cylinder basically appears 2°–8°after top dead centre (TDC). A double peak appears at

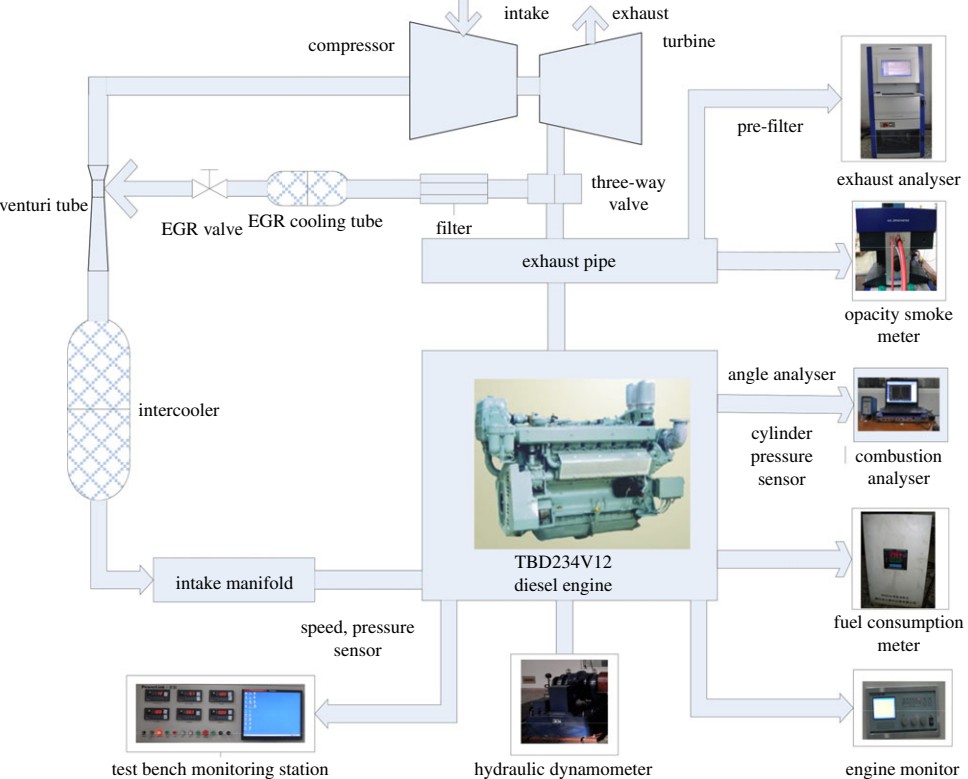

**Figure 3.** EGR system structure diagram.

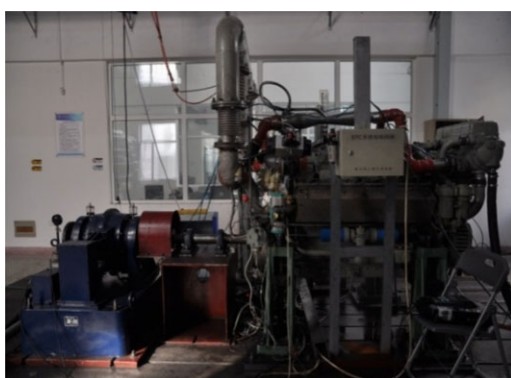

**Figure 4.** Test bench physical map.

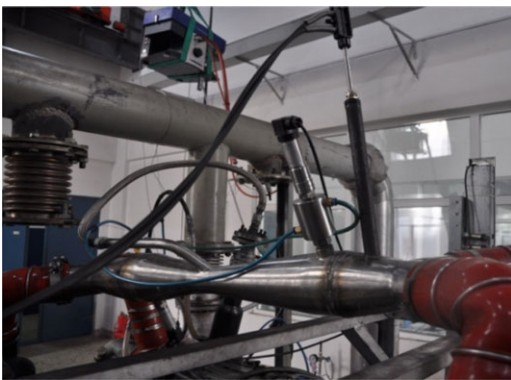

**Figure 5.** Venturi tube installation diagram.

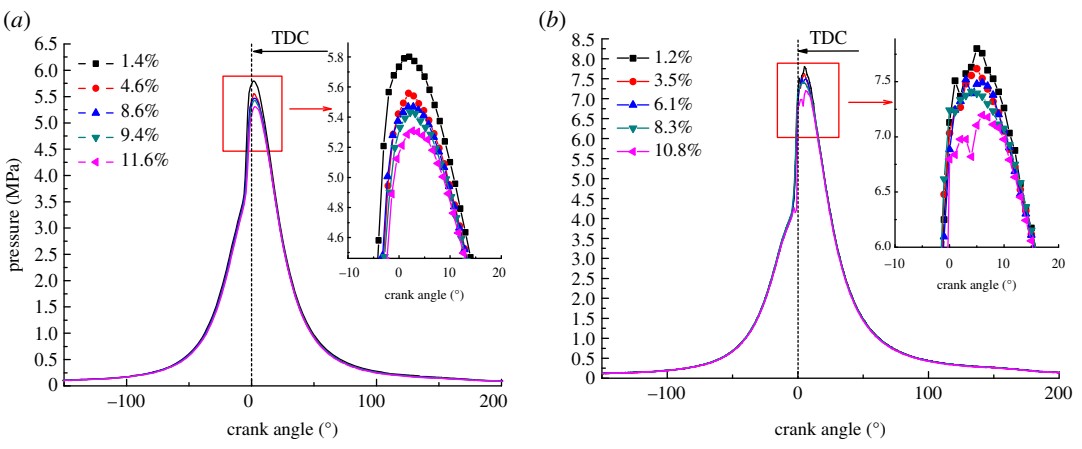

**Figure 6.** Effect of different EGR rates on cylinder pressure. (*a*) 900 r.p.m. and (*b*) 1500 r.p.m.

**Table 3.** Test schemes.

| operating point | engine speed (r.p.m.) | load percentage (%) | working time (min) |
|---|---|---|---|
| 1 | 900 | 25 | 10 |
| 2 | 900 | 50 | 10 |
| 3 | 900 | 75 | 10 |
| 4 | 1200 | 25 | 10 |
| 5 | 1200 | 50 | 10 |
| 6 | 1200 | 75 | 10 |
| 7 | 1500 | 25 | 10 |
| 8 | 1500 | 50 | 10 |
| 9 | 1500 | 75 | 10 |

high speed and the combustion pressure wave crests generally appear after TDC. With the increase in EGR rate, the pressure peak in cylinder decreases. As shown in figure 6*a*, at speed of 900 r.p.m., 25% load condition, the pressure peak decreased by 0.63 MPa. The reason is that the intake pressure and the intake volume reduced due to the introduction of the EGR exhaust gas. At the same time, the oxygen concentration in the mixture gas decreases while the number of inert gas molecules increases, the retardation effect on combustion increases, which leads to the decrease in initial pressure and peak pressure in the cylinder.

As shown in figure 6*b*, at 1500 r.p.m., 25% load condition, it can be seen that the cylinder pressure is bimodal and the first peak appears at 2° after TDC, the combustion occurs before TDC, mainly due to the upward movement of the piston. When the EGR rate increases, the first peak gradually decreases, and the second peak appears around 8° after TDC. When the piston reaches TDC, it starts to descend and the cylinder pressure decreases. However, the pressure increase caused by the combustion is greater than the pressure drop caused by the piston downward, so the second peak appears. When the EGR rate increases, the exhaust gas entering the cylinder increases and the specific heat capacity in cylinder rises, as a result, the ignition delay period becomes longer and the combustion starting point is delayed. Therefore, the introduction of EGR will cause the in-cylinder pressure curve to skew towards ATDC.

## 4.2. Influence of EGR rate on fuel consumption

Figure 7 shows the variation curve of fuel consumption rate with EGR rate when the diesel engine is at 900 and 1500 r.p.m. conditions, where the load is 25%, 50% and 75%. As can be seen from figure 7, the fuel consumption changes basically linearly with the EGR rate. This is due to the increase in exhaust volume, resulting in inadequate diesel combustion. When the diesel engine is at small load condition,

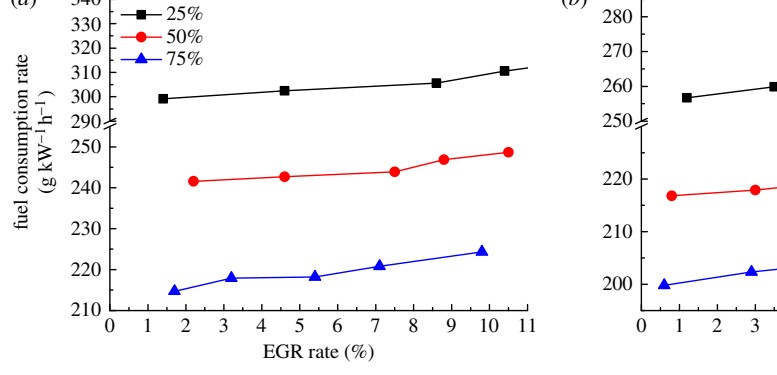

**Figure 7.** Effect of different EGR rates on fuel consumption. (a) 900 r.p.m. and (b) 1500 r.p.m.

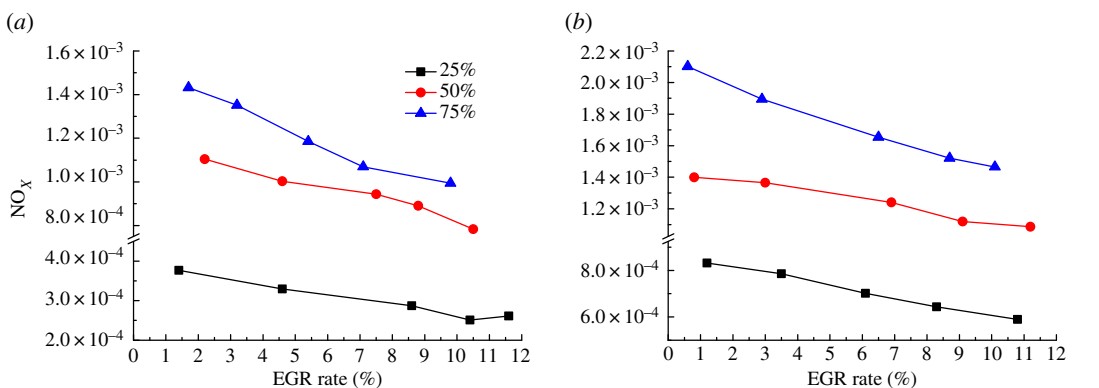

**Figure 8.** Effect of different EGR rates on NO$_X$. (a) 900 r.p.m. and (b) 1500 r.p.m.

the air–fuel ratio is large and the fresh air entering the cylinder is rich. The increase in the exhaust gas has little effect on the fuel consumption rate, so the fuel consumption rate changes slowly. For example, when the diesel engine is at 1500 r.p.m. speed, 25% load condition, when the EGR rate increases from 0 to 10.8%, the fuel consumption rate increases from 257.5 to 268.1 g kW$^{-1}$ h$^{-1}$, an increase of 4.12%.

However, when the diesel engine is at the high-load condition, the air combustion ratio is seriously reduced due to the introduction of EGR, resulting in excessive concentration of local mixed air in the cylinder and hypoxia. Therefore, with the increase in the EGR rate, the mixture concentration will continue to increase, which will eventually lead to combustion deterioration in the cylinder. In order to make up for the loss of power, sufficient power output is maintained by increasing the fuel injection quantity, so the fuel consumption rate increases with the increase in EGR rate. For example, at 1500 r.p.m. speed, 75% load condition, when the EGR rate increased from 0 to 10.8%, the fuel consumption rate increased from 200.4 to 212.2 g kW$^{-1}$ h$^{-1}$, with an increase of 5.89%. As the torque loss in the low-load area is small, the increase in oil consumption is more gentle than that in the high-load area.

## 4.3. Influence of EGR rate on NO$_X$ emissions

Figure 8 shows the variation curve of NO$_X$ emissions with EGR rate when the diesel engine is at 900 and 1500 r.p.m. conditions, where the load is 25%, 50% and 75%, respectively. It can be seen from figure 8 that the introduction of EGR exhaust gas can effectively improve NO$_X$ emissions, especially at high-load condition. When the EGR rate rises to around 8%, the NO$_X$ emissions can be reduced by about 25% on average. When the diesel engine is at 900 r.p.m. speed, 25% load condition, with the increase in the EGR rate, NO$_X$ emissions began to decrease from $3.7699 \times 10^{-4}$ to $2.512 \times 10^{-4}$. However, when the EGR rate continues to increase from 10.4 to 11.6%, the NO$_X$ increases by $1.01 \times 10^{-5}$. The main reason is that when the diesel engine is at low-speed condition, the EGR exhaust gas will reduce the overall oxygen concentration in the cylinder, which will also increase the temperature in the cylinder and the corresponding high-temperature duration will increase. Therefore, the reduction in oxygen concentration is offset, resulting in a negative effect of EGR on NO$_X$ emission.

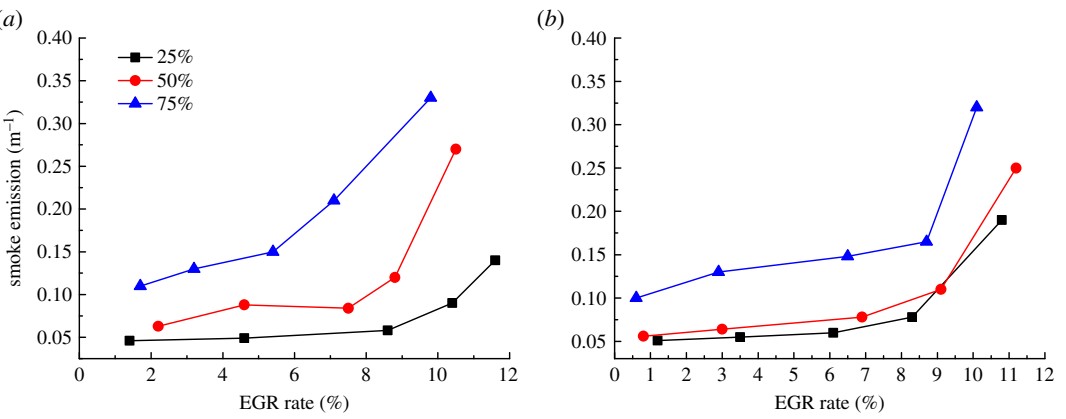

**Figure 9.** Effect of different EGR rates on smoke. (*a*) 900 r.p.m. and (*b*) 1500 r.p.m.

When the diesel engine speed increases to 1500 r.p.m., a large number of inert gases in the exhaust gas begin to play a prominent role that hinders the combustion, and the temperature in the cylinder decreases accordingly. As the generation rate of NO is lower than the reaction rate of combustion, only a small amount of NO is produced in the outer edge of the flame. With the increase in diesel speed, the duration of high temperature is shortened and the NO cannot reach the equilibrium content. Therefore, the production of $NO_X$ decreases with the increase in exhaust gas. For example, at 1500 r.p.m. and 75% load conditions, when the EGR rate changes from 0.6 to 10.1%, $NO_X$ decreases from $2.101 \times 10^{-4}$ to $1.465 \times 10^{-4}$, a total decrease of 30.27%.

## 4.4. Influence of EGR rate on smoke emission

Figure 9 shows the variation curve of smoke with EGR rate when the diesel engine is at 900 and 1500 r.p.m. conditions, where the load is 25%, 50% and 75%. As can be seen from figure 9, the change trend of smoke is basically opposite to that of $NO_X$ emissions. When the diesel engine is at low-load condition, the change in smoke is small. With the increase in load and EGR rate, the smoke emission increases. The high-smoke area was transferred to the high-load and high EGR rate area when the diesel engine is under the same speed condition. The main factors of the variation are the air–fuel ratio and the local temperature in the cylinder. For example, for 900 r.p.m., 25% load condition, the initial smoke emission remains unchanged. When the EGR rate increases to about 9%, the smoke rises significantly. With the constant increase of EGR rate, the smoke emission rises exponentially. Compared with the low-load condition, the break point of smoke under high-load condition was advanced accordingly. For example, when the diesel engine is at 900 r.p.m., 75% load conditions, the smoke emission increased sharply when the EGR rate reached 5.4%. This is mainly because when the diesel engine is under the same speed condition, the higher the load, the smaller the air combustion ratio and the more sensitive of EGR rate on the air combustion ratio. At the same time, the rise of temperature in the cylinder aggravates the secondary combustion, resulting in higher smoke emission. Therefore, the increase in smoke emissions is more obvious under high-load condition and excessive EGR rates should be avoided.

# 5. Performance assessment optimization analysis

The experimental results show that different EGR rates have different effects on diesel engines. Due to the limitations of this test, there is a certain loss in the process of mixing the intake and exhaust, and the maximum EGR rate is also limited to 15% and the data obtained are also limited. Considering the unique advantages of multi-objective grey decision-making theory in solving the problem of little data decision-making [25], an optimized grey decision-making method is proposed. In order to save space, the basic grey decision-making model is not described in detail.

## 5.1. Decision-making model optimization

It can be seen from the basic model [26] that the decision-making target and the corresponding target weight are the main factors. Considering the comprehensive influence of EGR rate on power, economy and emission performance, the fuel consumption rate, CO, $NO_X$, smoke and in-cylinder

pressure are selected as decision-making targets in this paper. As each decision-making target represents different aspects of diesel engine performance, the optimization decision-making problem between different performance is transformed into the weight problem of decision-making target.

The main purpose of EGR is to reduce the $NO_X$ emission, and the decision-maker always expect that the $NO_X$ emission can be reduced as low as possible without considering other factors. These 'expectations' should be transformed into the decision-making model so as to satisfy the actual operating requirements of EGR. Therefore, the method of subjective empowerment based on expert experience is used to assign $NO_X$ weight first according to different load conditions of the diesel engine. The specific principles are as follows: when the diesel engine is under low-load condition (load percentage $\leq 25\%$), the $NO_X$ emission concentration is low and it is suitable to adopt a lower EGR rate, thus make the $NO_X$ weight $\eta_3 = 0.3$. When the diesel engine is under high-load condition (load percentage $\geq 75\%$), the $NO_X$ emission concentration is high and it is suitable to adopt a higher EGR rate, thus make $NO_X$ weight $\eta_3 = 0.5$. When the diesel engine is under medium-load condition (25% < load percentage < 75%), the $NO_X$ concentration increased gradually and it is suitable to improve the EGR rate, thus make $NO_X$ weight $\eta_3 = 0.4$. In this way, it can not only meet the EGR characteristics under different operating conditions, but also avoids the deviations from the actual situation caused by objective empowerment.

The optimum EGR rate should reduce the $NO_X$ effectively while minimizing the negative impact on other diesel engines performances, and this requirement cannot be defined by specific criteria. Therefore, this paper tries to use the objective optimization method to solve the distribution of all decision-making targets from the perspective of data mining. The specific optimization process is as follows:

*Step 1:* To solve the optimal comprehensive distance. The optimized consistent effect measure matrix can be solved according to the optimized effect sample matrix

$$r'_{ij} = \begin{bmatrix} r_{11}^1 & r_{12}^1 & \cdots & r_{1m}^1 \\ r_{21}^2 & r_{22}^2 & \cdots & r_{2m}^2 \\ \cdots & \cdots & \cdots & \cdots \\ r_{n1}^k & r_{n2}^k & \cdots & r_{nm}^k \end{bmatrix}. \tag{5.1}$$

The best and worst values of different decision-making targets are defined: $p^{(k)+} = \max r_{1m}^{(k)}$ and $n^{(k)-} = \min r_{1m}^{(k)}$. The distance between each measure and the best value can be processed: $d^+(\eta_k) = |r_{nm}^k - p^{(k)+}|$, and the total distance of each strategy's consistent effect measure with best value can be expressed as a function of decision-making target weight: $D^+(\eta_k) = \sum_{i=1}^{n} \sum_{k=1}^{s} d^+(\eta_k)\eta_k$. In the same way, $d^+(\eta_k)$ and $D^-(\eta_k)$ can be obtained: $d^-(\eta_k) = |r_{nm}^k - n^{(k)-}|$, $D^-(\eta_k) = \sum\limits_{i=1}^{m} \sum\limits_{k=1}^{s} d^-(\eta_k)\eta_k$. According to the optimal distance principle, the optimization formula of comprehensive distance can be obtained

$$(\overline{O}) = \left\{ \max D^-(\eta_k) + \min D^+(\eta_k), (\text{s.t.} \sum_{k=1}^{s} \eta_k = 1, \eta_k \geq 0, k = 1, 2, \cdots, s) \right\}. \tag{5.2}$$

*Step 2:* To analyse the uncertainty of decision-making target weights using the grey entropy theory. The grey entropy of $\eta_k$ is defined at first [23]: $H_\otimes(\eta) = -\sum_{i=1}^{s} \eta_i \ln \eta_i$ and the maximum entropy can be obtained [27]

$$(\overline{O'}) = \left\{ \max H_\otimes(\eta) = -\sum_{i=1}^{s} \eta_i \ln \eta_i, \text{s.t.} \sum_{k=1}^{s} \eta_k = 1, \eta_k \geq 0, k = 1, 2, \cdots, s \right\}. \tag{5.3}$$

*Step 3:* To solve the optimal weight using the Lagrangian method. $(\overline{O})$ and $(\overline{O'})$ can be combined as follows:

$$\max \left\{ D^-(\eta_k) - D^+(\eta_k) + H_\otimes(\eta), \text{s.t.} \sum_{k=1}^{s} \eta_k = 1, \eta_k \geq 0, k = 1, 2, \cdots, s; u = \frac{1}{3} \right\}, \tag{5.4}$$

thus, the Lagrangian function of $\eta_k$ can be obtained and the $\eta_k$ can be solved

$$L(\eta_i, \lambda) = D^-(\eta_k) - D^+(\eta_k) + H_\otimes(\eta) + \lambda \left( \sum_{k=1}^{s} \eta_k - 1 \right) \tag{5.5}$$

and

$$\eta_k = \exp\{[\lambda + \mu d^-(\eta_k) - \mu d^+(\eta_k)]/(1 - 2\mu) - 1\}. \tag{5.6}$$

Since the weight of $NO_X$ has been assigned, only the other four decision-making targets are considered regardless of $NO_X$ when constructing the effect measure matrix. Knowing $\eta_3$ and $\eta_k$ ($k = 1, 2, 4, 5$), the final synthetic optimization weight $\eta'_k$ can be obtained and the optimized integrated effect measure $R$ is obtained according to formulae (5.1) and (5.2)

$$\eta'_k = (1 - \eta_3) \cdot \eta_k (k = 1, 2, 3, 4, 5). \tag{5.7}$$

## 5.2. Revision of assessment results

Since the objective decision-making model focuses on data mining more than the physical meaning of the data themselves, sometimes the assessment results will deviate from the actual situation. Therefore, in order to ensure the assessment results satisfy the actual EGR operating characteristics, the grey relational analysis algorithm [28,29] is adopted to optimize the decision-making result. The theoretical optimal value of each decision-making target is determined according to its effect measures. If the effect measure belongs to the upper effect measure, the maximum value corresponding to each EGR rate under this target is selected as the optimal value. If the effect measure belongs to the lower effect measure, the minimum value corresponding to each EGR rate under this target is selected as the optimal value. The theoretical optimal value of all decision-making targets constitutes the optimal sequence, i.e. the mother sequence

$$X_{\max} = \{x_{\max,1}, x_{\max,2}, \cdots, x_{\max,k}\}. \tag{5.8}$$

Each decision-making target value corresponding to different EGR rates constitute the sub-sequence

$$X_i = \{x_1, x_2, \cdots, x_k\}, \tag{5.9}$$

where $i$ represents the different EGR rate and $k$ represents different decision targets. As a result, the correlation coefficient between sub-sequence and optimal sequence can be obtained

$$r_i = \frac{1}{N} \sum_{k=1}^{N} \left( \frac{\min\limits_{i} \min\limits_{k} |x_0(k) - x_i(k)| + \rho \max\limits_{i} \max\limits_{k} |x_0(k) - x_i(k)|}{|x_0(k) - x_i(k)| + \rho \max\limits_{i} \max\limits_{k} |x_0(k) - x_i(k)|} \right), \tag{5.10}$$

where $\rho$ is the resolution ratio $a$, $x_0(k)$ is the parent sequence and $x_i(k)$ is the sub-sequence. The higher the correlation coefficient, the closer to the optimal sequence, i.e. the comprehensive performance of the diesel engine under the current EGR rate is closest to the theoretical optimal performance. At last, the optimized integrated effect measure can be obtained by multiplying the correlation coefficient $r_i$ by the corresponding integrated effect measure $R$

$$R' = R \cdot r_i. \tag{5.11}$$

## 5.3. Optimal EGR rate

Part of test conditions are selected for verification analysis. The specific data are shown in table 4.

OP1, OP2 represent 25% load condition under different speed. OP3, OP4 represent 50% load condition under different speed. OP5, OP6 represent 75% load condition under different speed. Taking OP1 as example and it can be obtained according to (5.1) and (5.2).

The same effect measure matrix $r_{ij}^{(k)}$

$$(r_{ij}^{(5)}) = \begin{bmatrix} 1.0000 & 0.9891 & 0.9791 & 0.9633 & 0.9556 \\ 0.9154 & 0.9552 & 0.9712 & 0.9775 & 1.0000 \\ 0.6663 & 0.7620 & 0.8743 & 1.0000 & 0.9613 \\ 1.0000 & 0.9388 & 0.7931 & 0.5111 & 0.3286 \\ 1.0000 & 0.9482 & 0.8508 & 0.6663 & 0.5306 \end{bmatrix}$$

The optimized weight vector $\eta'$

$$\eta' = [0.1364 \quad 0.1456 \quad 0.3000 \quad 0.2258 \quad 0.1922].$$

The integrated effect measure $R$

$$R = [0.8876 \quad 0.8968 \quad 0.8798 \quad 0.8172 \quad 0.7405].$$

**Table 4.** Part of the operating point test data.

| operating point (OP) | EGR rate | fuel consumption | CO | NO$_x$ | smoke | cylinder pressure |
|---|---|---|---|---|---|---|
| OP1 | 2.4 | 299.2 | 336.37 | 376.99 | 0.046 | 5.8001 |
| | 4.6 | 302.5 | 354.73 | 329.65 | 0.049 | 5.5583 |
| | 8.6 | 305.6 | 395.36 | 287.32 | 0.058 | 5.4671 |
| | 10.4 | 310.6 | 504.82 | 251.2 | 0.09 | 5.4318 |
| | 11.6 | 313.1 | 633.94 | 231.3 | 0.14 | 5.3094 |
| OP2 | 2.2 | 256.7 | 196.2 | 832.45 | 0.051 | 7.7996 |
| | 4.5 | 259.9 | 211.3 | 786.3 | 0.055 | 7.6176 |
| | 7.1 | 261.3 | 229.4 | 701.6 | 0.06 | 7.5171 |
| | 9.3 | 266.8 | 273.5 | 643.5 | 0.078 | 7.4105 |
| | 11.8 | 268.1 | 380.6 | 589.7 | 0.19 | 7.1965 |
| OP3 | 2.2 | 241.6 | 316.57 | 1104.5 | 0.063 | 7.2545 |
| | 4.6 | 242.7 | 335.53 | 1002.6 | 0.088 | 7.2108 |
| | 7.5 | 243.9 | 366.7 | 943.5 | 0.084 | 7.1393 |
| | 9.8 | 246.9 | 427.84 | 890.65 | 0.12 | 7.0167 |
| | 11.5 | 248.7 | 503.62 | 783.6 | 0.27 | 6.9568 |
| OP4 | 1.8 | 216.8 | 164.3 | 1399.8 | 0.056 | 9.0835 |
| | 4 | 217.9 | 169.9 | 1365.4 | 0.064 | 9.01549 |
| | 7.9 | 220.8 | 176.9 | 1240.6 | 0.078 | 8.8452 |
| | 9.1 | 224.5 | 190.4 | 1119 | 0.11 | 8.7263 |
| | 11.2 | 226.1 | 280.5 | 1086.4 | 0.25 | 8.5216 |
| OP5 | 1.7 | 214.7 | 286.53 | 1432 | 0.11 | 9.1568 |
| | 4.2 | 217.9 | 304.76 | 1351.6 | 0.13 | 9.0763 |
| | 7.4 | 218.2 | 329.89 | 1185 | 0.15 | 8.8016 |
| | 9.1 | 220.8 | 366.54 | 1069.4 | 0.21 | 8.7569 |
| | 11.8 | 224.3 | 426.71 | 994.2 | 0.33 | 8.5597 |
| OP6 | 1.6 | 199.8 | 156.4 | 2101 | 0.1 | 10.5505 |
| | 3.9 | 202.3 | 164.2 | 1894 | 0.13 | 10.4165 |
| | 7.5 | 205 | 172.2 | 1653 | 0.148 | 10.2256 |
| | 9.7 | 209.2 | 206 | 1521 | 0.165 | 10.0584 |
| | 11.1 | 212.2 | 312.3 | 1465 | 0.32 | 9.8568 |

The associated sequence is shown in table 5.

The correlation coefficient $r_i$

$$r_i = [0.9138 \quad 0.9373 \quad 0.9223 \quad 0.8877 \quad 0.9079]$$

The optimized integrated effect measure

$$R' = [0.8111 \quad 0.8206 \quad 0.8115 \quad 0.7254 \quad 0.6723]$$

It can be seen from the result that the performance ranking of each EGR rate under OP1 operating conditions is: 4.6, 8.6, 2.4, 10.4 and 11.6%, i.e. the optimal EGR rate is 4.6%.

In the same way, the evaluation results of other operating points can be obtained as shown in table 6.

As can be seen from the evaluation results in table 6, when the diesel engine is at the low-load conditions, the smaller EGR rate achieved a relatively higher assessment value, as the EGR rate

**Table 5.** Associated sequence.

| $X_{max}$ | 299.2 | 336.37 | 251.2 | 0.046 | 5.3094 |
|---|---|---|---|---|---|
| X1 | 299.2 | 336.37 | 376.99 | 0.046 | 5.8001 |
| X2 | 302.5 | 354.73 | 329.65 | 0.049 | 5.5583 |
| X3 | 305.6 | 395.36 | 287.32 | 0.058 | 5.4671 |
| X4 | 310.6 | 504.82 | 251.2 | 0.09 | 5.4318 |
| X5 | 313.1 | 633.94 | 231.3 | 0.14 | 5.3094 |

**Table 6.** Other evaluation results.

| OP | optimized integrated effect measure $R'$ | performance ranking | optimal EGR rate (%) |
|---|---|---|---|
| OP2 | [0.8222 0.8372 0.8144 0.7609 0.6265] | 4.5%, 2.2%, 7.1%, 9.3%, 11.8% | 4.5 |
| OP3 | [0.7992 0.8052 0.8258 0.7796 0.7423] | 7.5%, 4.6%, 2.2%, 9.8%, 11.5% | 7.5 |
| OP4 | [0.8380 0.8390 0.8417 0.8275 0.7365] | 7.9%, 4%, 1.8%, 9.1%, 11.2% | 7.9 |
| OP5 | [0.7610 0.7978 0.8086 0.8180 0.8057] | 9.1%, 7.4%, 11.8%, 4.2%, 1.7% | 9.1 |
| OP6 | [0.7641 0.8034 0.8351 0.8452 0.7839] | 9.7%, 7.5%, 3.9%, 11.1%, 1.6% | 9.7 |

increases, the corresponding assessment value decreases and the decline is obvious, especially when the EGR is high, resulting in the highest EGR rate achieving the minimum value. For example, for OP2 condition, when the EGR rate rises to 7.1%, the optimized integrated effect measure value decreases obviously. When the diesel engine is at the medium-load conditions, the optimized integrated effect measure value increases as the EGR rate increases. However, when the EGR rate is high, the evaluation value decreases gradually. For example, for OP3 condition, when the EGR rate rises to 9.8%, the optimized integrated effect measure value decreases obviously. When the diesel engine is at the high-load conditions, the lower EGR rate achieved the smaller optimized integrated effect measure value, with the increase in EGR rate, the optimized integrated effect measure value increases linearly. However, when the EGR rate is too high, the evaluation value obviously decreases instead. For example, for OP5 and OP6 condition, when the EGR rate rises to 11.8% and 11.1%, both the optimized integrated effect measure values decrease obviously, and compared to OP5, the decline of OP6 is more obvious. In conclusion, a small EGR rate should be used when the diesel engine is in low-load condition. As the load increases, the content of $NO_X$ increases and the EGR rate should increase moderately. When the diesel engine is at high-load condition, a higher EGR rate should be used to reduce $NO_X$ effectively, but in order to avoid the negative impact of excessive exhaust gas on the performance of the diesel engine, the EGR rate should be controlled within a certain range.

## 6. Conclusion

(1) The venturi tube can meet the injection requirements of exhaust gas with which the high-pressure EGR mode can be realized successfully. When the diesel engine is running according to the load characteristics, the EGR can effectively reduce the $NO_X$ emissions and the reduction is more obvious, especially for high-load condition. The $NO_X$ emissions are significantly reduced with the increase in EGR rate. When the EGR rate increases to about 8%, the $NO_X$ emission can be reduced by about 25%.

(2) The increase in EGR rate results in different degrees of delay in the start and endpoints of combustion; the fuel consumption increases slightly when the diesel engine is at low-load condition. The increase in fuel consumption in the high-load condition is more obvious than that in the low-load condition when the diesel engine is at the same speed; the effect of EGR rate on smoke is not obvious for low-load condition. When the EGR rate increases to about 9%, the smoke basically increases exponentially with the increase in EGR rate. Compared with the low-load condition, the break point of smoke value under high load was advanced.

(3) The optimal EGR decision-making optimization method based on grey correlation coefficient modified can successfully realize the evaluation of EGR performance, and the final decision-making results are basically in line with the actual operating characteristics of diesel engine EGR. It can effectively weaken the subjectivity of the empirical methods and improve the rationality of the decision-making results, which provides a new research method for EGR optimization research.

Data accessibility. Data available from the Dryad Digital Repository at: https://doi.org/10.5061/dryad.451c4s2 [30].
Authors' contributions. X.-h.Z., C.-l.Y. and H.-C.W. are responsible for test design and implementation, X.-h.Z. and H.-C.W. are responsible for system installation. X.-h.Z., Y.-y.W. and C.-l.Y. are responsible for results analysis. All authors gave final approval for publication.
Competing interests. We declare we have no competing interests.
Funding. Financial support came from Marine Low-Speed Engine Project-Phase I (grant no. CDG01-KT0302).
Acknowledgements. The authors thank for the support of the fund, thanks for the help of everybody in the experiment, thanks to the reviewers!!

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
