## [Reviewer comments · Royal Society Open Science]

Review History

RSOS-181907.R0 (Original submission)

Review form: Reviewer 1

Is the manuscript scientifically sound in its present form?

Yes

Are the interpretations and conclusions justified by the results?

Yes

Is the language acceptable?

No

Is it clear how to access all supporting data?

Yes

Do you have any ethical concerns with this paper?

No

Have you any concerns about statistical analyses in this paper?

No

Recommendation?

Accept with minor revision (please list in comments)

Comments to the Author(s)

The topic of this paper is very distinctive and I'm personally interested in it. In this paper, the authors have done a meaningful work about EGR system design and performance test, and an optimal EGR decision-making optimization method based on grey correlation coefficient modified is firstly proposed. The case studies have illustrated that the venture tube EGR system achieve good results and the method proposed by the authors can effectively solve the EGR performance evaluation problem, which has great potential application value for EGR performance optimization and diesel engine performance improvement. In general, the manuscript is well organized and I recommend to be accepted after minor revised. There are some comments on your paper after review:

1. The written English is suggested to be modified to avoid grammatical errors.
2. The authors need to add the measurement precision of the test instrument, and it is suggest to check the manuscript carefully to avoid spelling mistakes such as 'NOX'.
3. The method proposed by the authors is rarely used in diesel engine field and can solve the practical problem of performance optimization, the final result also proves the feasibility of the method. If it can be extended to the performance optimization of automobile engine, the practical value of the method will be greatly improved.

Review form: Reviewer 2 (A. Kalaiselvane)

Is the manuscript scientifically sound in its present form?

Yes

Are the interpretations and conclusions justified by the results?

Yes

Is the language acceptable?

Yes

Is it clear how to access all supporting data?

Yes

Do you have any ethical concerns with this paper?

No

Have you any concerns about statistical analyses in this paper?

No

Recommendation?

Accept with minor revision (please list in comments)

Comments to the Author(s)

The authors should explain in detail the design of Venturi Tube EGR System with calculations, citing the references.

Decision letter (RSOS-181907.R0)

15-Mar-2019

Dear Dr Zu,

The editors assigned to your paper ("Experimental study on diesel engine EGR performance and Optimum EGR Rate Determination Method") have now received comments from reviewers. We would like you to revise your paper in accordance with the referee and Associate Editor suggestions which can be found below (not including confidential reports to the Editor). Please note this decision does not guarantee eventual acceptance.

Please submit a copy of your revised paper before 07-Apr-2019. Please note that the revision deadline will expire at 00.00am on this date. If we do not hear from you within this time then it will be assumed that the paper has been withdrawn. In exceptional circumstances, extensions may be possible if agreed with the Editorial Office in advance. We do not allow multiple rounds of revision so we urge you to make every effort to fully address all of the comments at this stage. If deemed necessary by the Editors, your manuscript will be sent back to one or more of the original reviewers for assessment. If the original reviewers are not available, we may invite new reviewers.

- Data accessibility

If you wish to submit your supporting data or code to Dryad (<http://datadryad.org/>), or modify your current submission to dryad, please use the following link:
<http://datadryad.org/submit?journalID=RSOS&manu=RSOS-181907>

- **Competing interests**

- **Authors' contributions**

- **Acknowledgements**

- **Funding statement**

Kind regards,

Royal Society Open Science Editorial Office
Royal Society Open Science
openscience@royalsociety.org

on behalf of Professor R. Kerry Rowe (Subject Editor)
openscience@royalsociety.org

Associate Editor comments:

Reviewer 1 has recommended that the English within the manuscript will need to be polished. A number of language polishing services are available for authors whose first language is not English. We recommend that you ask a native speaker of English or solicit the support of a language polishing service (<https://royalsociety.org/journals/authors/language-polishing/>) prior to resubmitting the manuscript.

Comments to Author:

Reviewers' Comments to Author:

Reviewer: 1

Comments to the Author(s)

The topic of this paper is very distinctive and I'm personally interested in it. In this paper, the authors have done a meaningful work about EGR system design and performance test, and an optimal EGR decision-making optimization method based on grey correlation coefficient modified is firstly proposed. The case studies have illustrated that the venture tube EGR system achieve good results and the method proposed by the authors can effectively solve the EGR performance evaluation problem, which has great potential application value for EGR performance optimization and diesel engine performance improvement. In general, the manuscript is well organized and I recommend to be accepted after minor revised. There are some comments on your paper after review:

1. The written English is suggested to be modified to avoid grammatical errors.
2. The authors need to add the measurement precision of the test instrument, and it is suggest to check the manuscript carefully to avoid spelling mistakes such as 'NOX'.
3. The method proposed by the authors is rarely used in diesel engine field and can solve the practical problem of performance optimization, the final result also proves the feasibility of the method. If it can be extended to the performance optimization of automobile engine, the practical value of the method will be greatly improved.

Reviewer: 2

Comments to the Author(s)

The authors should explain in detail the design of Venturi Tube EGR System with calculations, citing the references.

Author's Response to Decision Letter for (RSOS-181907.R0)

See Appendix A.

RSOS-181907.R1 (Revision)

Review form: Reviewer 1

Is the manuscript scientifically sound in its present form?

Yes

Are the interpretations and conclusions justified by the results?

Yes

Is the language acceptable?

Yes

Is it clear how to access all supporting data?

Yes

Do you have any ethical concerns with this paper?

No

Have you any concerns about statistical analyses in this paper?

No

Recommendation?

Accept as is

Comments to the Author(s)

The article has been revised well and it is recommended for acceptance.

Decision letter (RSOS-181907.R1)

10-May-2019

Dear Dr Zu,

I am pleased to inform you that your manuscript entitled "Experimental study on diesel engine EGR performance and Optimum EGR Rate Determination Method" is now accepted for publication in Royal Society Open Science.

Kind regards,

Andrew Dunn

on behalf of Prof R. Kerry Rowe (Subject Editor)

Associate Editor Comments to Author:

After further review and editor-consideration, we are of the view the manuscript may be accepted in its current form.

Reviewer comments to Author:

Reviewer: 1

Comments to the Author(s)

The article has been revised well and it is recommended for acceptance.

Appendix A

Dear Editors and Reviewers:

Thank you for your letter and for the reviewers' comments concerning my manuscript entitled "**Experimental study on diesel engine EGR performance and Optimum EGR Rate Determination Method**" (*ID: RSOS-181907*).

I am very grateful to your valuable advice and the comments are all helpful for revising and improving my paper, as well as the important guiding significance to my researches. I have studied comments carefully and have made correction which we hope meet with approval. Revised portion are marked in red in the paper. The main corrections in the paper and the responds to the reviewer's comments are as flowing:

Reviewer #1

Comment #1: The written English is suggested to be modified to avoid grammatical errors..

Response:

Thank you very much for your comments.

I am very sorry for my incorrect writing and the written English is modified carefully by an expert.

Comment #2: The authors need to add the measurement precision of the test instrument, and it is suggest to check the manuscript carefully to avoid spelling mistakes such as 'NOX'..

Response:

Thank you very much for your comments.

The measurement precision of the test instrument has been added and the spelling mistakes has been modified carefully.

Comment #3: The method proposed by the authors is rarely used in diesel engine field and can solve the practical problem of performance optimization, the final result also proves the feasibility of the method. If it can be extended to the performance optimization of automobile engine, the practical value of the method will be greatly improved.

Response:

Thank you very much for your comments.

First of all, I agree with the expert's opinion. However, due to the different characteristics of EGR requirements for vehicle engines and marine engines and the method in this paper is based on test data. If it is required to be applied to a vehicle engine, it is necessary to provide test data for the vehicle engine, while this paper is mainly for marine engines. Since there is no test data for the vehicle engine, there is no way to implement it currently. However, I believe that the research ideas in this paper can provide a certain reference for vehicle engines.

Reviewer #2

Comment #1: The authors should explain in detail the design of Venturi Tube EGR System with calculations, citing the references.

Response:

Thank you very much for your comments. The design and calculation of Venturi Tube EGR

System has been explained as follows:

The main parameters that affecting the performance of the venturi are the throat area and the cone angle of the diffuser. The throat area determines the ejector capacity of the venture and the cone angle of the diffuser determines the recovery of the gas pressure after mixing. To simplify the calculation, the flow is considered to be a constant flow situation, and the gas dynamics formula is applied^[23]:

$$\begin{array}{ll} \text{Equation of state for ideal gas:} & p=\rho RT \\ & (1) \end{array}$$

$$\begin{array}{ll} \text{Flow continuous equation} & m=\rho Av \\ & (2) \end{array}$$

$$\begin{array}{ll} \text{Intake sound velocity equation} & a=\sqrt{\gamma RT} \\ & (3) \end{array}$$

$$\begin{array}{ll} \text{Mach number calculation equation} & M = \frac{v}{a} \\ & (4) \end{array}$$

Where p —intake pressure, MPa.

ρ —intake density, kg/m³.

R —gas constant, J/(kg•K).

T —the absolute temperature of the intake air, K.

m —intake air mass flow, kg/s.

A —pipe cross-sectional area, m².

v —intake air flow rate, m/s ;

a —local speed of sound, m/s.

γ —specific heat ratio.

M —Mach number.

In this paper, the rated working condition of diesel engine (1800r/min, 444kW) was selected as the venturi tube design condition. The selection of this operating point is based on the following considerations: The flow in the pipe is simplified to constant flow, when the diesel engine is working in the design working condition and the opening degree of the EGR valve is adjusted from fully closed to fully open, the venturi can be started. By the action of the pressure-reducing ejector, the EGR rate required can be achieved, and when the EGR valve is fully opened, the throat portion of the venturi tube will not occluded. Due to the limitation of the original machine structure, the medium-cold high-pressure exhaust gas circulation system is selected in this test.

The basic boundary conditions are determined as follows: : the diameter of the inlet and outlet of the venturi tube should be equal to the diameter of the intake pipe $d_1 = d_2 = 115\text{mm}$, the air pressure at the outlet of the compressor $p_1 = 0.157\text{MPa}$, the temperature $T_1 = 345\text{K}$, the flow rate of the intake air $m_1 = 0.529\text{kg/s}$, the pressure of the exhaust gas before the turbine $p_2 = 0.15\text{MPa}$, According to the gas dynamics equation, it can be calculated:

$$\text{Intake density: } \rho = \frac{p_1}{RT_1} = \frac{1.57 \times 10^5}{287.04 \times 345} = 1.5854 \text{ kg/m}^3 ;$$

$$\text{Intake flow rate: } v_1 = \frac{m_1}{\rho A_1} = \frac{0.529}{1.5854 \times 0.115^2 \times 3.14/4} = 32.124 \text{ m/s} ;$$

$$\text{Local sound speed: } a_1 = \sqrt{\gamma RT_1} = \sqrt{1.4 \times 287.04 \times 345} = 372.344 \text{ m/s} ;$$

$$\text{Mach number: } M_1 = \frac{v_1}{a_1} = \frac{32.124}{372.344} = 0.08627 ;$$

According to the Mach number, linear interpolation is used to check the gas dynamic function table:

$$\frac{A_1}{A_*} = 6.7598 \quad \frac{p_1}{p_0} = 0.9948 \quad d_* = 44 \text{ mm}$$

Where: A_* —critical section area, m^2 ; p_0 —stagnation pressure, MPa.

The magnitude of the pre-turbine exhaust pressure determines the design value of the venturi throat pressure. In order to achieve a good ejector effect, it is necessary to form a certain pressure difference between the exhaust pipe and the throat portion of the venturi. According to empirical data, a pressure of 3 to 10 KPa is generally required^[24], as a result:

$$p_t = (0.15 - 0.01) \times 10^6 = 0.14 \text{ Mpa}$$

$$\frac{p_1}{p_t} = \frac{0.157}{0.14} = 1.1214$$

$$\frac{p_t}{p_0} = \frac{p_1/p_0}{p_1/p_t} = \frac{0.9948}{1.1214} = 0.8871$$

According to the Mach number, linear interpolation is used to check the gas dynamic function table:

$$M_2 = 0.41722, \quad \frac{A_t}{A_*} = 1.5372$$

$$d_t = \sqrt{\frac{A_t}{\pi/4}} = \sqrt{\frac{A_t}{\pi/4} \cdot \frac{A_t/A_*}{A_t/A_{cr}}} = \sqrt{\frac{1.5372}{6.7598}} \times 0.115 = 54.84 \text{ mm} > d_*$$

Therefore, the design meets the requirements. After determining the throat area, the nozzle length of the venturi L_1 , the length of the mixing section L_t and the length of the diffuser L_2 are determined according to the empirical formula. In order to balance the space arrangement of the test bench, the selection of the total length L must be feasible.

In this test, the shrinkage cone angle $\alpha = 24^\circ$ which meet the empirical value $10^\circ < \alpha < 40^\circ$.

$$L_1 = \frac{d_1 - d_t}{2 \text{tg}(\alpha/2)} = \frac{0.115 - 0.05484}{2 \text{tg} 12^\circ} = 0.1415 \text{ m} = 141.5 \text{ mm}$$

Considering the overall size layout of the test bench, set $L = 441.5 \text{ mm}$, $L_t = 50 \text{ mm}$

$$L_2 = L - L_1 - L_t = 400 - 141.5 - 50 = 245 \text{ mm}$$

Therefore, the diffuser angle:

$$\beta = 2 \arctan \frac{d_1 - d_t}{2 \times L_2} = 2 \arctan \frac{0.115 - 0.05484}{2 \times 245} = 14^\circ$$

According to experience, the diffuser angle β should be within the range $11^\circ < \beta < 18^\circ$ ^[25], so the designed diffuser section meets the requirements. The total calculated parameters are shown in table 2.

Table 2. Venturi tube main parameters

The throat area	54.84mm	The length of the mixing section	50mm
Shrink cone angle	24°	The length of the diffuser	245mm
The length of the venturi nozzle	141.5mm	The diffuser angle	14°
The total length	441.5mm		

Yours
Sincerely
Xiang-huan Zu